# Phosphorylations of serines 21/9 in glycogen synthase kinase 3α/β are dispensable for $^{V600E}$BRAF-driven premalignant tumour development in the mouse intestine

Pooyeh Farahmand[1¤], Paulina Rzasa[1], Caleb Green[1], Fiona Hey[1], Susan Giblett[1], Hong Jin[1], Kevin West[2], Nicolas B. Sylvius[3], Catrin A. Pritchard[1*], Alessandro Rufini[1,4*]

**1** Leicester Cancer Research Centre, University of Leicester, Clinical Sciences Building, Leicester, United Kingdom, **2** Department of Cellular Pathology, University Hospitals of Leicester, Leicester, United Kingdom, **3** NUCLEUS Genomics, Core Biotechnology Services, University of Leicester, Leicester, United Kingdom, **4** Dipartimento di Bioscience, University of Milan, Milan, Italy

¤ Current address: University of Glasgow/Cancer Research UK Scotland Institute, Glasgow, UK
* cap8@le.ac.uk (CAP), alessandro.rufini@unimi.it (AR)

## Abstract

Valine to glutamate substitution at residue 600 of the *BRAF* oncogene ($^{V600E}$BRAF mutation) is prevalent in human colorectal cancers with a serrated histopathology and is thought to be a founder mutation. Using a conditional knock-in mouse model we have previously demonstrated that $^{V600E}$Braf drives crypt hyperplasia in the short term as well as shortened survival linked to increased tumour burden in the long-term. These phenotypes are associated with induction of gene signatures for E2F targets, MYC signalling, G2/M transition, canonical Wnt signalling, and cholesterol biosynthesis. Although these gene signatures are reverted by MEK inhibition, there remains a lack of understanding of the signalling pathways involved, particularly the mechanism of crosstalk between the MAPK and Wnt pathways. Here, we have examined a role for phosphorylation of GSK3αβ isoforms at residues S21/S9. By introducing homozygous knock-in mutations for *Gsk3αβ*S9A/S21A onto the $^{V600E}$Braf background, we unexpectedly show a marginal effect of these mutations on further increasing crypt proliferation. However, this impact is lost in the long-term as there are no significant differences in mouse survival, tumour burden or tumour grade. Consistently, the *Gsk3αβ* knock-in mutations do not change the transcriptional programme induced by $^{V600E}$Braf, except for 3 genes (*Ephx4*, *Eif2b3 Ppp1r13l*) whose expression is significantly altered, potentially contributing to the short-term increase in crypt hyperplasia. Overall, our data show that therapeutic strategies targeting *GSK3αβ* phosphorylation at serines 21/9 are not worthwhile options for $^{V600E}$BRAF colorectal cancers.

**Data availability statement:** All microarray data files are available from the GEO database and accession numbers have been provided in the manuscript's methods section. Original uncropped western blots have also been provided as Supplemental data.

**Funding:** This study was financially supported by the Cancer Research UK Programme (https://www.cancerresearchuk.org) in the form of a grant (A13083) received by CAP. This study was also financially supported by the Cancer Prevention Research Trust (https://www.cancerpreventionresearch.co.uk) in the form of a grant (RM60G0817) received by AR. This study was also financially supported by the Medical Research Council (https://www.ukri.org/councils/mrc) in the form of a grant (MR/N013913/1). The funders had no role in study design, data collection and analysis, decision to publish, or preparation of the manuscript.

**Competing interests:** The authors have declared that no competing interests exist.

## Introduction

Serrated colorectal cancers (CRCs) account for ~10% of CRC cases and arise from sessile serrated precursor lesions with a distinct flat and mucinous morphology on the right side of the colon [1,2]. The most common driver and founder oncogenic mutation detected in serrated CRCs is the $^{V600E}BRAF$ mutation [3]. Additional molecular characteristics associated with serrated CRCs include microsatellite instability (MSI), CpG island methylator phenotype (CIMP) and mutations of the Transforming Growth Factor (TGF) signalling pathway [1,2,4–6]

To understand how $^{V600E}BRAF$ drives CRC tumorigenesis, conditional knock-in mouse models have been extensively utilised [5–9]. These studies show that $^{V600E}Braf$ is a weak oncogene when expressed on its own in the mouse intestine, inducing crypt hyperplasia in the short term and premalignant lesions after long latency and showing limited ability for malignant progression. Cooperation with other mutations such as components of the TGFβor Wnt signalling pathways enhance the carcinogenesis process [5,6,8]. We have also recently demonstrated that expression of $^{V600E}Braf$ induces transcriptional reprogramming with the enrichment of gene signatures indicative of proliferation, Wnt signalling and metabolic reprogramming, particularly a cholesterol metabolism signature, and the negative enrichment of an intestinal differentiation signature [9,10].

The presence of $^{V600E}Braf$ in the mouse intestine induces constitutive activation of the MEK/ERK pathway and crypt hyperplasia, and transcriptional patterns, including Wnt signatures, that can be reversed by MEK inhibition [9]. However, the link between Wnt signalling and $^{V600E}Braf$-driven tumour development remains uncertain, and it is not presently clear as to whether this involves direct crosstalk of the signalling pathways or cooperation via mutational activation/inactivation mechanisms. There have been several reported mechanisms of crosstalk between MAPK and Wnt pathways for example via intermediary kinases such as p90[RSK] or p70[S6K] [11] that converge at the level of the glycogen synthase kinase-3 (GSK3)αβ, a key component of the destruction complex that regulates Wnt pathway activation [12,13]. The two isoforms of GSK3αβ (GSK3α and GSK3β) are encoded by different genes. Despite their high similarity, the two kinases are not always functionally redundant. Indeed, loss of *Gsk3β* is not compatible with life due to liver toxicity and cardiomyopathy, whereas mice with a homozygous knockout of *Gsk3α* are viable, though affected by impaired spermatogenesis [14–16]. Regarding the regulation of the Wnt pathway, experiments using genetic deletions of the two isoforms have shown redundancy in the regulation of the Wnt pathway [17,18]. However, *Gsk3β* has a major role in the regulation of β-catenin and selective pharmacological inhibition of GSK3β triggers β-catenin stabilization and activation of the Wnt pathway, with no evident compensation from GSK3α [19]. Both pro- and anti-tumorigenic functions have been documented for GSK3αβ and the multifunctional nature of these serine/threonine kinases is reflected in the fact that they have over 100 substrates and interacting molecules [20]. GSK3αβ is normally active in cells but is inactivated in response to intracellular/extracellular stimuli, in part by N-terminal inhibitory phosphorylation at residues S21/S9 [17,21]. Disruption of phosphorylation of GSK3αβ at these sites has been linked

to multiple pathologies including cancer [20] and, in the context of $^{V600E}Braf$ expression in the gastrointestinal tract, strong induction of GSK3α/β serines 21/9 phosphorylation has been detected [7].

To investigate the functional relevance of GSK3αβ N-terminal phosphorylation in the context of $^{V600E}Braf$-driven tumour development, we have analysed the intestinal phenotype of $^{V600E}Braf$ knock-in mice following intercrossing with mice bearing homozygous knock-ins of $Gsk3αβ$ serines 21/9 to alanines. We show that these phosphorylation events have no effect on mouse survival, tumour burden or tumour grade induced by $^{V600E}Braf$. A marginal impact on crypt hyperplasia following short-term induction of $^{V600E}Braf$ is observed but this is associated with the changes in expression of only 3 of the 1962 genes altered by expression of $^{V600E}Braf$ alone.

## Materials and methods

### Experimental animals

Animal experiments were performed according to Home Office guidelines under Home Office project license P7B8067BB. The study protocol was discussed and approved by the University of Leicester local ethics committee. $Braf^{LSL-V600E/+}$ mice [7,22] and $VillinCreER^{T+/0}$ mice and the interbreeding of these animals has been reported previously [9]. The $Gsk3α^{S21A/+}$ and $Gsk3β^{S9A/+}$ mice used for this study have been previously reported [23,24] and were provided by Dario Alessi, MRC Phosphorylation Unit, Dundee, UK. All mice used for breeding were maintained on the C57BL6 background in standard housing conditions and aged between 8 weeks and 6 months. 64 animals were generated for this study (Table 2). Male and female mice were randomly selected for all experiments. All mice were maintained as previously detailed [9] and genotyping of $Braf^{LSL-V600E/+}$ and $VillinCreER^{T+/0}$ mice by PCR of genomic DNA isolated from ear samples was also undertaken as reported in this publication. Genotyping of $Gsk3α^{S21A/+}$ and $Gsk3β^{S9A/+}$ mice by PCR of genomic DNA isolated from ear samples was undertaken as described [23,24]. Induction of Cre activity in experimental mice was achieved by injection of 1 mg Tamoxifen (TM) into the abdominal cavity of mice aged 8–12 weeks on 3 consecutive days at 24-hour intervals. TM was prepared as a 10 mg/ml stock solution solubilised in corn oil. For proliferation analysis, mice were injected with 1 mg of 5-bromo-2'-deoxyuridine (BrdU) by intraperitoneal injection and harvested 3 hours later [7]. For the survival data and analysis of tumour development, animals were kept on study and were monitored routinely until they became moribund after which they were sacrificed humanely according to Home Office regulations. In all cases, survival time was represented as the duration between the last TM injection and the termination day. For the short-term studies, mice were sacrificed 72 hours after the last TM injection. Animals were inspected daily for loss of body condition and/or signs of significant suffering. Mice were humanely killed immediately by cervical dislocation if displaying two or more of the following: Body Condition Score <3/5 [25], weight loss of 15–20% of the animal's starting weight (or from maximum weight if adult), hunched posture, piloerection, under-activity, failure to eat/drink normally over 24–48 hours, dehydration, altered respiratory rate, pallor, tremors, abnormal vocalisation. If any one of these endpoints was deemed severe, the animals were culled immediately. Mice were also immediately humanely killed if showing any of the following signs: weight loss of 20% of the animal's starting weight (or from maximum weight if adult), Body Conditions Score <2/5, internal tumour growth begins impeding normal function, serious clinical signs such as diarrhoea or dyspnoea, pain, which cannot be promptly relieved, palpation of abdominal tumours >10 mm diameter. Tumour growth was monitored by palpation twice weekly. After harvesting of the intestine, "macroscopic tumours" were those defined as visible by eye. Microscopic tumours not visible by eye were detectable through microscopic analysis of intestinal tissue. All personnel involved in animal handling underwent Home Office-approved training (personal license) and additional local training by authorised personnel was provided for animal handling, culling and procedures implemented in the study protocol.

### Histology and protein analysis

At end point, the entire intestine was removed and flushed with PBS. The small intestine was removed and cut into 6 equal segments while the large intestine was processed in its entirety. A small piece of each segment (~0.5 cm) was

taken and frozen in liquid nitrogen for protein extraction or snap frozen in RNAlater (Sigma) for RNA extraction. The remainder of each segment was cut longitudinally and "swiss-rolled" as described previously [7]. Each roll was placed in 4% (w/v) paraformaldehyde (PFA) in PBS and rocked gently for 16–18 hours. The PFA was decanted, and the tissue was washed and stored in 70% (v/v) ethanol at 4°C. Tissue was processed and embedded in paraffin for sectioning. 5 μm sections were cut using a microtome and then used for H&E staining or immunostaining as previously described [7,9,23]. Primary antibodies used for immunostaining were: BrdU (Cell signalling, 5292, 1:500), phospho-histone H3 (Cell signalling, 9701, 1:500) and β-catenin (Sigma, Cat No. C2206, 1:2000). Brightfield scans of stained tissue were captured at 40x magnification using Nikon 3 microscope and associated ND acquisition software, with the help of the advanced imaging facility (AIF) at the University of Leicester. Stained images were visualised with a Leica light micro-scope. For counting of stained tissue, only well-orientated crypts were counted and a minimum of 50 crypts per animal was scored for each experiment. All counting and measurements were undertaken blind and conducted on at least 3 animals per genotype.

Protein lysates from small intestine samples were prepared as previously reported and subjected to western blot analysis [23]. Primary antibodies used were: phospho-GSK3αβ (Ser21/9) (Cell Signalling, 9331) or ERK2 (Santa Cruz, sc-1647).

## Transcriptomic analysis

Tissue from samples stored in RNAlater was extracted on a Promega Maxwell 16 using the Maxwell 16 LEV sim-plyRNA Cells Kit. Each experimental group consisted of RNA from 3 males and 3 females of the same genotype. RNA quality and quantity was assessed using the 2100 Bioanalyzer Instrument (Agilent). Transcriptome profiling was conducted using SurePrint G3 Mouse Gene Expression v2 8x60K Microarray Kit (Agilent, #G4852B) following manu-facturer's instructions.

## Bioinformatic analysis

**Microarray dataset.** Differential expression analysis on microarray data was performed in R (v 4.3.2) using the limma R Bioconductor package (v3.46.0) [26]. The dataset was normexp background corrected followed by quantile normalization. The linear modelling approach and the empirical Bayes statistics were implemented using the lmFit and eBayes functions, respectively. The contrast matrix was customized to compare KI/KI/BVE/Cre genotype to WT/WT/WT/Cre, KI/KI/WT/Cre and WT/WT/BVE/Cre separately. For each of three comparisons, genes showing BH-adjusted p-value < 0.05 as well as p-value < 0.2 were retrieved with topTable function. Common genes in all three comparisons were considered significantly altered in KI/KI/BVE/Cre genotype. Principal component analysis (PCA) was assessed with prcomp function from the R stats package (v4.3.2) and visualized using ggplot2 package (v3.4.4). Gene Set Variation Analysis (GSVA) [27] method was performed to investigate differences in Wnt gene signature [28] activity and Wnt pathway expression (KEGG) among all genotypes. GSVA enrichment scores for each sample were calculated using gsva function with default settings from GSVA R package (v1.52.3). limma package was used to perform a differential expression analysis at pathway level. For visualisation, the ggplot2 and pheatmap (v1.0.12) (https://CRAN.R-project.org/package=pheatmap) packages were used to generate boxplots and heatmaps, respectively. Adjusted p-values were added to boxplots using stat_pvalue_manual function from ggpubr package (v0.6.0) (https://rpkgs.datanovia.com/ggpubr/).

## TCGA dataset

TCGA-COAD STAR count matrices were retrieved with TCGAbiolinks R package (v2.32.0) and analysed with DESeq2 package (v1.44.0) [29,30]. Boxplots of selected genes were generated using ggplot2 package. BH-adjusted p-values were added using stat_pvalue_manual function from ggpubr package.

## Statistical analysis

For pairwise comparisons, significance was determined by t-test or Wilcoxon matched pairs test. For quantification of β-catenin, statistical analysis was performed using One-Way ANOVA with Tukey's multiple comparison test. Survival analysis was assessed via Logrank test. Graphs were produced in GraphPad Prism (v10.2.3), plotted as mean ± SD, and p values <0.05 were considered significant.

## Results

### Generation of *Gsk3αS21A/S21A; Gsk3βS9A/S9A; BrafLSL-V600E/+; VilCreERT+/0* mice

Mice homozygous for the combined *Gsk3α^S21A^* and *Gsk3β^S9A^* mutations were generated as previously reported [23]. Four rounds of breeding were then undertaken to introduce the homozygous *Gsk3αβ* mutations onto the *Braf^LSL-V600E/+^; VilCreER^T+/0^* background (S1A Fig). In the first round of breeding, *Gsk3α^S21A/S21A^; Gsk3β^S9A/S9A^* mice were inter-crossed with *VilCreER^T+/0^* mice. The resulting offspring from this cross are shown in Table 1 (Mating 1). In parallel, *Gsk3α^S21A/S21A^; Gsk3β^S9A/S9A^* mice were intercrossed with *Braf^LSL-V600E/+^* mice and the resulting offspring are shown in Table 1 (Mating 2). At the next round of breeding, *Gsk3α^S21A/+^; Gsk3β^S9A/+^; VilCreER^T+/0^* mice from Mating 1 were intercrossed with *Gsk3α^S21A/+^; Gsk3β^S9A/+^; Braf^LSL-V600E/+^* mice from Mating 2. The resulting offspring are shown in Table 1 (Mating 3). These included *Gsk3α^S21A/S21A^; Gsk3β^S9A/S9A^; Braf^LSL-V600E/+^; VilCreER^T+/0^* mice and relevant controls. Subsequently, in Mating 4 (Table 2), *Gsk3α^S21A/S21A^; Gsk3β^S9A/S9A^; Braf^LSL-V600E/+^; VilCreER^T+/+^* mice from Mating 3 were intercrossed with *Gsk3α^S21A/S21A^; Gsk3β^S9A/S9A^; Braf^+/+^; VilCreER^T+/0^* mice from Mating 3 to obtain mice homozygous for the *Gsk3αβ* knock-in alleles while also containing the inducible *Braf^LSL-V600E^* allele. The frequency of obtained versus expected genotypes from Mating 4 is shown in Table 2 and indicates that the combined genotypes do not affect the frequency of live births (p = 0.54, chi-squared test).

### Effect of *Gsk3αβ* mutations on survival and tumour development following *V600EBraf* induction

First, we confirmed that the presence of the *Gsk3α^S21A/S21A^; Gsk3β^S9A/S9A^* knock-in mutations abolished phosphorylation of GSK3αβ on the *^V600E^Braf* background in the intestinal tissue (S1B Fig). To study the effect of genotype on survival, *Gsk3α^S21A/S21A^; Gsk3β^S9A/S9A^; Braf^+/+^; VilCreER^T+/0^* (KI/KI/WT/Cre) mice and *Gsk3α^S21A/S21A^; Gsk3β^S9A/S9A^; Braf^LSL-V600E/+^; VilCreER^T+/0^* (KI/KI/BVE/Cre) mice from Mating 4 were treated with tamoxifen to induce expression of *^V600E^Braf* in the mouse intestine as previously reported [9]. Simultaneously, control mice with the genotypes *Gsk3α^+/+^; Gsk3β^+/+^; Braf^+/+^; VilCreER^T+/0^* (WT/WT/WT/Cre) and *Gsk3α^+/+^; Gsk3β^+/+^; Braf^LSL-V600E/+^; VilCreER^T+/0^* (WT/WT/BVE/Cre) were also treated with tamoxifen. Survival of the mice of each of the 4 genotypes is shown in Fig 1A and the comparison of Mean Survival Times (MST) is shown in Fig 1B. As previously reported [9], there was a significant reduction in MST as a result of *^V600E^Braf* expression in WT/WT/BVE/Cre mice in comparison to the tamoxifen-treated *VilCreER^T+/0^* control (WT/WT/WT/Cre). We have also previously reported that the presence of the *Gsk3αβ* mutations has no significant effect on MST of mice on the wild-type background [23] and, consistent with this, there was no statistically significant difference in the MST of WT/WT/WT/Cre compared to KI/KI/WT/Cre mice. In addition, no significant difference in MST was observed upon comparing WT/WT/BVE/Cre and KI/KI/BVE/Cre mice. Thus, *^V600E^Braf* expression in the intestine leads to a significant reduction in MST, which is not further influenced by the *Gsk3αβ* knock-in mutations (Fig 1A, B).

We have previously reported the detection of macroscopic intestinal tumours in *^V600E^Braf*-expressing mice at endpoint [7,9]. Similarly, in the present study, tumours from the small intestine were visible to the naked eye (Fig 1C, top image) in tissue harvested from both WT/WT/BVE/Cre and KI/KI/BVE/Cre mice, whereas no visible tumours were detectable in the small intestine of control mice (WT/WT/WT/Cre or KI/KI/WT/Cre) or in colon tissue. Upon analysis of H&E-stained sections, tumours showed a consistent appearance across WT/WT/BVE/Cre and KI/KI/BVE/Cre mice (Fig 1C, bottom images). In most tumours, the muscle layer was intact, and the structures of the villi and crypts were recognisable, although there was an increase in mitotic figures. Tumour burden was quantified across mice of each genotype. However,

**Table 1. Initial first three rounds of mouse breeding to introduce *Gsk3* knock-in mutations onto the *Braf*$^{LSL-V600E/+}$; *VilCreER*$^{T+/0}$ background.**

**Mating 1:** *VilCreER*$^{T+/0}$ X *Gsk3α*$^{S21A/S21A}$; *Gsk3β*$^{S9A/S9A}$
Genotypes of offspring:

*Gsk3α*$^{S21A/+}$; *Gsk3β*$^{S9A/+}$; *VilCreER*$^{T+/+}$

*Gsk3α*$^{S21A/+}$; *Gsk3β*$^{S9A/+}$; *VilCreER*$^{T+/0}$

**Mating 2:** *Braf*$^{LSL-V600E/+}$ X *Gsk3α*$^{S21A/S21A}$; *Gsk3β*$^{S9A/S9A}$
Genotypes of offspring:

*Gsk3α*$^{S21A/+}$; *Gsk3β*$^{S9A/+}$; *Braf*$^{LSL-V600E/+}$

*Gsk3α*$^{S21A/+}$; *Gsk3β*$^{S9A/+}$; *Braf*$^{+/+}$

**Mating 3:** *Gsk3α*$^{S21A/+}$; *Gsk3β*$^{S9A/+}$; *Braf*$^{LSL-V600E/+}$ X *Gsk3α*$^{S21A/+}$; *Gsk3β*$^{S9A/+}$; *VilCreER*$^{T+/0}$
Genotypes of expected offspring:

*Gsk3α*$^{S21A/21A}$; *Gsk3β*$^{S9A/S9A}$; *Braf*$^{LSL-V600E/+}$; *VilCreER*$^{T+/0}$

*Gsk3α*$^{S21A/21A}$; *Gsk3β*$^{S9A/S9A}$; *Braf*$^{LSL-V600E/+}$; *VilCreER*$^{T+/+}$

*Gsk3α*$^{S21A/21A}$; *Gsk3β*$^{S9A/S9A}$; *Braf*$^{+/+}$; *VilCreER*$^{T+/0}$

*Gsk3α*$^{S21A/21A}$; *Gsk3β*$^{S9A/S9A}$; *Braf*$^{+/+}$; *VilCreER*$^{T+/+}$

*Gsk3α*$^{S21A/21A}$; *Gsk3β*$^{S9A/+}$; *Braf*$^{LSL-V600E/+}$; *VilCreER*$^{T+/0}$

*Gsk3α*$^{S21A/21A}$; *Gsk3β*$^{S9A/+}$; *Braf*$^{LSL-V600E/+}$; *VilCreER*$^{T+/+}$

*Gsk3α*$^{S21A/21A}$; *Gsk3β*$^{S9A/+}$; *Braf*$^{+/+}$; *VilCreER*$^{T+/0}$

*Gsk3α*$^{S21A/21A}$; *Gsk3β*$^{S9A/+}$; *Braf*$^{+/+}$; *VilCreER*$^{T+/+}$

*Gsk3α*$^{S21A/21A}$; *Gsk3β*$^{+/+}$; *Braf*$^{LSL-V600E/+}$; *VilCreER*$^{T+/0}$

*Gsk3α*$^{S21A/21A}$; *Gsk3β*$^{+/+}$; *Braf*$^{LSL-V600E/+}$; *VilCreER*$^{T+/+}$

*Gsk3α*$^{S21A/21A}$; *Gsk3β*$^{+/+}$; *Braf*$^{+/+}$; *VilCreER*$^{T+/0}$

*Gsk3α*$^{S21A/21A}$; *Gsk3β*$^{+/+}$; *Braf*$^{+/+}$; *VilCreER*$^{T+/+}$

*Gsk3α*$^{S21A/+}$; *Gsk3β*$^{S9A/S9A}$; *Braf*$^{LSL-V600E/+}$; *VilCreER*$^{T+/0}$

*Gsk3α*$^{S21A/+}$; *Gsk3β*$^{S9A/S9A}$; *Braf*$^{LSL-V600E/+}$; *VilCreER*$^{T+/+}$

*Gsk3α*$^{S21A/+}$; *Gsk3β*$^{S9A/S9A}$; *Braf*$^{+/+}$; *VilCreER*$^{T+/0}$

*Gsk3α*$^{S21A/+}$; *Gsk3β*$^{S9A/S9A}$; *Braf*$^{+/+}$; *VilCreER*$^{T+/+}$

*Gsk3α*$^{S21A/+}$; *Gsk3β*$^{S9A/+}$; *Braf*$^{LSL-V600E/+}$; *VilCreER*$^{T+/0}$

*Gsk3α*$^{S21A/+}$; *Gsk3β*$^{S9A/+}$; *Braf*$^{LSL-V600E/+}$; *VilCreER*$^{T+/+}$

*Gsk3α*$^{S21A/+}$; *Gsk3β*$^{S9A/+}$; *Braf*$^{+/+}$; *VilCreER*$^{T+/0}$

*Gsk3α*$^{S21A/+}$; *Gsk3β*$^{S9A/+}$; *Braf*$^{+/+}$; *VilCreER*$^{T+/+}$

*Gsk3α*$^{S21A/+}$; *Gsk3β*$^{+/+}$; *Braf*$^{LSL-V600E/+}$; *VilCreER*$^{T+/0}$

*Gsk3α*$^{S21A/+}$; *Gsk3β*$^{+/+}$; *Braf*$^{LSL-V600E/+}$; *VilCreER*$^{T+/+}$

*Gsk3α*$^{S21A/+}$; *Gsk3β*$^{+/+}$; *Braf*$^{+/+}$; *VilCreER*$^{T+/0}$

*Gsk3α*$^{S21A/+}$; *Gsk3β*$^{+/+}$; *Braf*$^{+/+}$; *VilCreER*$^{T+/+}$

*Gsk3α*$^{+/+}$; *Gsk3β*$^{S9A/S9A}$; *Braf*$^{LSL-V600E/+}$; *VilCreER*$^{T+/0}$

*Gsk3α*$^{+/+}$; *Gsk3β*$^{S9A/S9A}$; *Braf*$^{LSL-V600E/+}$; *VilCreER*$^{T+/+}$

*Gsk3α*$^{+/+}$; *Gsk3β*$^{S9A/S9A}$; *Braf*$^{+/+}$; *VilCreER*$^{T+/0}$

*Gsk3α*$^{+/+}$; *Gsk3β*$^{S9A/S9A}$; *Braf*$^{+/+}$; *VilCreER*$^{T+/+}$

*Gsk3α*$^{+/+}$; *Gsk3β*$^{S9A/+}$; *Braf*$^{LSL-V600E/+}$; *VilCreER*$^{T+/0}$

*Gsk3α*$^{+/+}$; *Gsk3β*$^{S9A/+}$; *Braf*$^{LSL-V600E/+}$; *VilCreER*$^{T+/+}$

*Gsk3α*$^{+/+}$; *Gsk3β*$^{S9A/+}$; *Braf*$^{+/+}$; *VilCreER*$^{T+/0}$

*Gsk3α*$^{+/+}$; *Gsk3β*$^{S9A/+}$; *Braf*$^{+/+}$; *VilCreER*$^{T+/+}$

*Gsk3α*$^{+/+}$; *Gsk3β*$^{+/+}$; *Braf*$^{LSL-V600E/+}$; *VilCreER*$^{T+/0}$

*Gsk3α*$^{+/+}$; *Gsk3β*$^{+/+}$; *Braf*$^{LSL-V600E/+}$; *VilCreER*$^{T+/+}$

*Gsk3α*$^{+/+}$; *Gsk3β*$^{+/+}$; *Braf*$^{+/+}$; *VilCreER*$^{T+/0}$

*Gsk3α*$^{+/+}$; *Gsk3β*$^{+/+}$; *Braf*$^{+/+}$; *VilCreER*$^{T+/+}$

**Table 2. Last round of mouse breeding to introduce *Gsk3* knock-in mutations onto the *Braf^{LSL-V600EI/+}*; *VilCreER^{T+/0}* background.**

| | Observed number (%) | Expected number (%) |
|---|---|---|
| **Mating 4:** $Gsk3\alpha^{S21A/S21A}$; $Gsk3\beta^{S9A/S9A}$; $Braf^{-SL-V600EI/+}$; $VilCreER^{T+/+}$ x $Gsk3\alpha^{S21A/S21A}$; $Gsk3\beta^{S9A/S9A}$; $Braf^{+/+}$; $VilCreER^{T+/0}$ Genotypes of offspring: | | |
| $Gsk3\alpha^{S21A/S21A}$; $Gsk3\beta^{S9A/S9A}$; $Braf^{-SL-V600EI/+}$; $VilCreER^{T+/+}$ | 18 (28.12%) | 16 (25%) |
| $Gsk3\alpha^{S21A/S21A}$; $Gsk3\beta^{S9A/S9A}$; $Braf^{+/+}$; $VilCreER^{T+/0}$ | 18 (28.12%) | 16 (25%) |
| $Gsk3\alpha^{S21A/S21A}$; $Gsk3\beta^{S9A/S9A}$; $Braf^{-SL-V600EI/+}$; $VilCreER^{T+/0}$ | 17 (26.56%) | 16 (25%) |
| $Gsk3\alpha^{S21A/S21A}$; $Gsk3\beta^{S9A/S9A}$; $Braf^{+/+}$; $VilCreER^{T+/+}$ | 11 (17.19%) | 16 (25%) |
| Total | **64 (100%)** | **64 (100%)** |

as shown in Fig 1D, the genotype of the mouse did not affect the number of visible tumours. Regression analysis was performed and showed that, as expected, age was significantly associated with the number of macroscopic tumours (p = 0.004) in both WT/WT/BVE/Cre and KI/KI/BVE/Cre mice. However, there was no significant difference in the number of tumours between the different genotypes after accounting for age (p = 0.4) (Fig 1D).

To investigate whether the *Gsk3αβ* knock-in mutations had an impact on tumour aggressiveness, intestinal tumours from WT/WT/BVE/Cre and KI/KI/BVE/Cre mice were subjected to pathological grading (Fig 1E). In low-grade dysplastic tumours (LGD), the muscle layers were found to be intact, and the structure of the villi were not affected although nucleus to cytoplasmic ratio changes were detectable in epithelial cells. In high-grade dysplastic (HGD) tumours, large nuclei were detectable with an abundance of mitotic figures and the structures of the crypt and villi were disrupted but still recognisable. Invasive carcinomas showed loss of cell polarity and infiltration of the tumours into the muscle layers (carcinoma). Quantitation of tumours of each grade in mice of each genotype showed a tendency towards an increase in the mean number of LGD and HGD tumours/mouse in mice with the *Gsk3αβ* mutations compensated by a lower mean number of carcinomas. However, statistical analysis using pairwise comparisons showed this was not significant (Fig 1E).

### Effect of *Gsk3αβ* mutations on crypt hyperplasia following *V600EBraf* induction

We have previously reported crypt hyperplasia following short-term expression of *V600EBraf* in the intestine that is reversed by inhibition of the MAPK pathway [7,9]. This phenotype has also been associated with increased phosphorylation of GSK3 at serines 9/21 [7], although GSK3 phosphorylation of these residues alone without *V600EBraf* expression does not affect crypt homeostasis [23]. To investigate if GSK3αβ phosphorylation at serine residues 9/21 is required for *V600EBraf*-induced crypt hyperplasia, we investigated crypt cell number in the WT/WT/WT/Cre, WT/WT/BVE/Cre and KI/KI/BVE/Cre mice derived above following treatment with TM for 3 days.

The intestines of WT/WT/BVE/Cre and KI/KI/BVE/Cre mice both demonstrated significantly increased crypt cell number following TM treatment for 3 days compared to WT/WT/WT/Cre controls (p = 0.0008 and p < 0.0011 respectively; Fig 2A). The combined knock-in mutations of *Gsk3αβ* with *V600EBraf* induction showed slightly greater crypt cell number compared to *V600EBraf* induction alone, but this was not significant (p = 0.1142; Fig 2A). To examine if there are alterations in crypt proliferation, we stained intestinal tissue with an antibody for the mitotic marker phospho-histone H3 (pH3), which is known to increase in S phase of the cell cycle, and quantified the number of stained cells/crypt. In this analysis, the expression of *V600EBraf* alone in WT/WT/BVE/Cre mice did not demonstrate significantly altered number of pH3 positive cells per crypt compared to control WT/WT/WT/Cre mice (p = 0.3693; Fig 2B). However, the combined knock-in mutations of *Gsk3αβ* with *V600EBraf* induction showed significantly greater number of pH3 positive cells/crypt compared to control mice (p = 0.0022; Fig 2B) and to mice expressing *V600EBraf* alone (p = 0.0142; Fig 2B). We also examined actively cycling cells by pre-treating

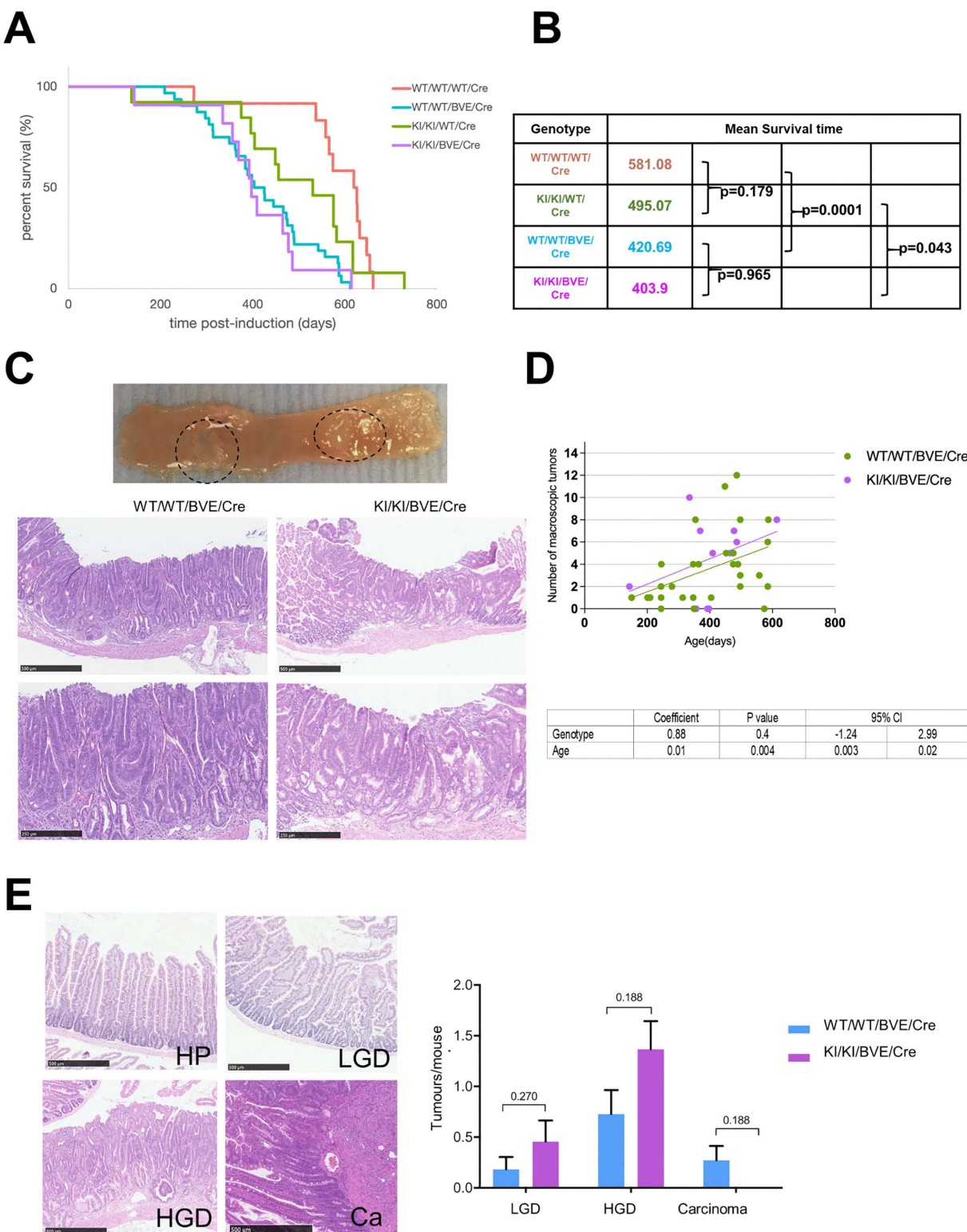

**Fig 1. Effect of genotype on survival and tumour development.** (A) Kaplan-Meier survival graph indicating the effect of genotype on overall survival. Mice were injected with TM and were monitored routinely for signs of poor health. Animals were terminated humanely following signs of ill health and age of death was recorded. Data were obtained from: WT/WT/WT/Cre (n = 11), WT/WT/BVE/Cre (n = 31), KI/KI/WT/Cre (n = 11), KI/KI/BVE/Cre (n = 11)

animals. (B) Table showing Mean Survival Time (MST) of mice with the four different combinations of mutations shown in (A) following treatment with TM. Statistical analysis was carried out to calculate significance of the data following comparison between each group (Logrank test). (C) Photograph of a section of small intestine indicating visible macroscopic tumours (black circles) is shown in the top image. Representative images of H&E-stained sections of tumours from WT/WT/BVE/Cre and KI/KI/BVE/Cre animals are shown at the bottom. Scale bars = 500 µm (5x magnification, top panels) and 250 µm (10x magnification lower panels). (D) Scatter graph of number of macroscopic tumours/mouse at endpoint in V600EBraf-expressing mice with and without Gsk3αβ knock-in mutations taking age of death into consideration. Data were obtained from 11 mice of each genotype. The table below shows statistical analysis of the effect of genotype and age on macroscopic tumour burden using linear regression model. (E) Gsk3αβ knock-in mutations have no effect on tumour grade. Each tumour from $^{V600E}$Braf-expressing mice was subjected to histopathological analysis of H&E-stained sections. The top left panel shows an example of hyperplastic crypts (HP). The top right panel shows an example of low-grade dysplastic (LGD) tissue. At this stage, the muscle layer is intact, and the structure of the villus is not affected but there are nucleus to cytoplasmic ratio changes in the epithelial cells. In the bottom left panel, an example of a high-grade dysplastic (HGD) tumour is shown indicating the presence of large nuclei, an abundance of mitotic figures and the structures of the crypt and villi are disrupted but still recognisable. On the bottom right panel, the image shows an invasive carcinoma in which there is loss of polarity and infiltration of the tumour into the muscle layer. Scale bars = 500 µm, 5x magnification. The graph on the right indicates the number of tumours per mouse at endpoint of each grade in $^{V600E}$ Braf-expressing mice with and without the Gsk3αβ knock-in mutations. Data shows mean number of tumours of each grade for 11 mice of each genotype and error bars indicate SD. Significance was determined by the Wilcoxon matched pairs test and p values are indicated.

mice with BrdU before harvesting of tissue. Staining and quantitation of BrdU positive cells in the crypts was then undertaken. This analysis showed that $^{V600E}$Braf expression in the intestine increased the number of cycling BrdU positive cells in the crypts compared to wild-type controls nearly reaching significance (p = 0.054; Fig 2C). The combined knock-in mutations of Gsk3αβ with $^{V600E}$Braf induction also showed significantly greater number of BrdU positive cells/crypt compared to wild-type control mice (p = 0.0086; Fig 2C). Although the number of BrdU positive cells/crypt were slightly increased in KI/KI/BVE/Cre mice compared to WT/WT/BVE/Cre mice following TM induction, this was not statistically significant (p = 0.44; Fig 2C). Overall, these data show that short-term $^{V600E}$Braf expression in the mouse intestine increases crypt cell number, with increased proliferation at least partially accounting for this increase. The presence of Gsk3αβ knock-in mutations on top of short-term $^{V600E}$Braf expression in the mouse intestine can exacerbate this phenomenon.

### Gene expression analysis

To investigate whether phosphorylation of GSK3αβ at serine residues 21/9 influences the transcriptional changes elicited by expression of $^{V600E}$Braf, we performed microarray analysis in small intestinal tissue isolated from controls and KI/KI mice. Principal component analysis of transcriptomic data indicates that variability among groups is principally driven by expression of mutant Braf, with little to no effect of the Gsk3αβ knock-in mutations (Fig 3A). We then pinpointed differentially expressed genes (DEGs), whose expression was significantly and selectively altered in KI/KI/BVE/Cre mice and identified only 3 genes (False Discovery Rate, FDR = 0.05) (Fig 3B, upper heatmap and Fig 3C), confirming that abolishing GSK3 N-terminal phosphorylation has a negligible impact on the gene expression programme orchestrated by $^{V600E}$Braf, a finding supported by the minimal impact on the number of DEGs produced by increasing the FDR to 0.2 (Fig 3B, bottom heatmap). The three genes altered in KI/KI/BVE/Cre mice are Epoxide Hydrolase 4 (Ephx4), showing statistically significant increased expression, and Eukaryotic Translation Initiation Factor 2B Subunit Gamma (Eif2b3) and Protein Phosphatase 1 Regulatory Subunit 13 Like (Ppp1r13l) with statistically significant decreased expression compared to the other genotypes. We also assessed expression of the same genes in human CRCs from the TCGA database [31] and found that all genes displayed significant increased expression in the tumour tissue (Fig 3D), although independently of BRAF mutational status.

Finally, we checked the activity of the Wnt pathway. We observed reduced nuclear β-catenin staining in KI/KI/BVE/Cre intestinal tissue compared to WT/WT/BVE/Cre intestinal tissue (Fig 4A). At the transcriptomic level, as previously shown [9], we observed a significant increase in expression of Wnt target genes [28] in mice expressing $^{V600E}$Braf in the intestine (Fig 4B) and a concomitant reduction in the expression of a Wnt pathway component gene signature comprising the main

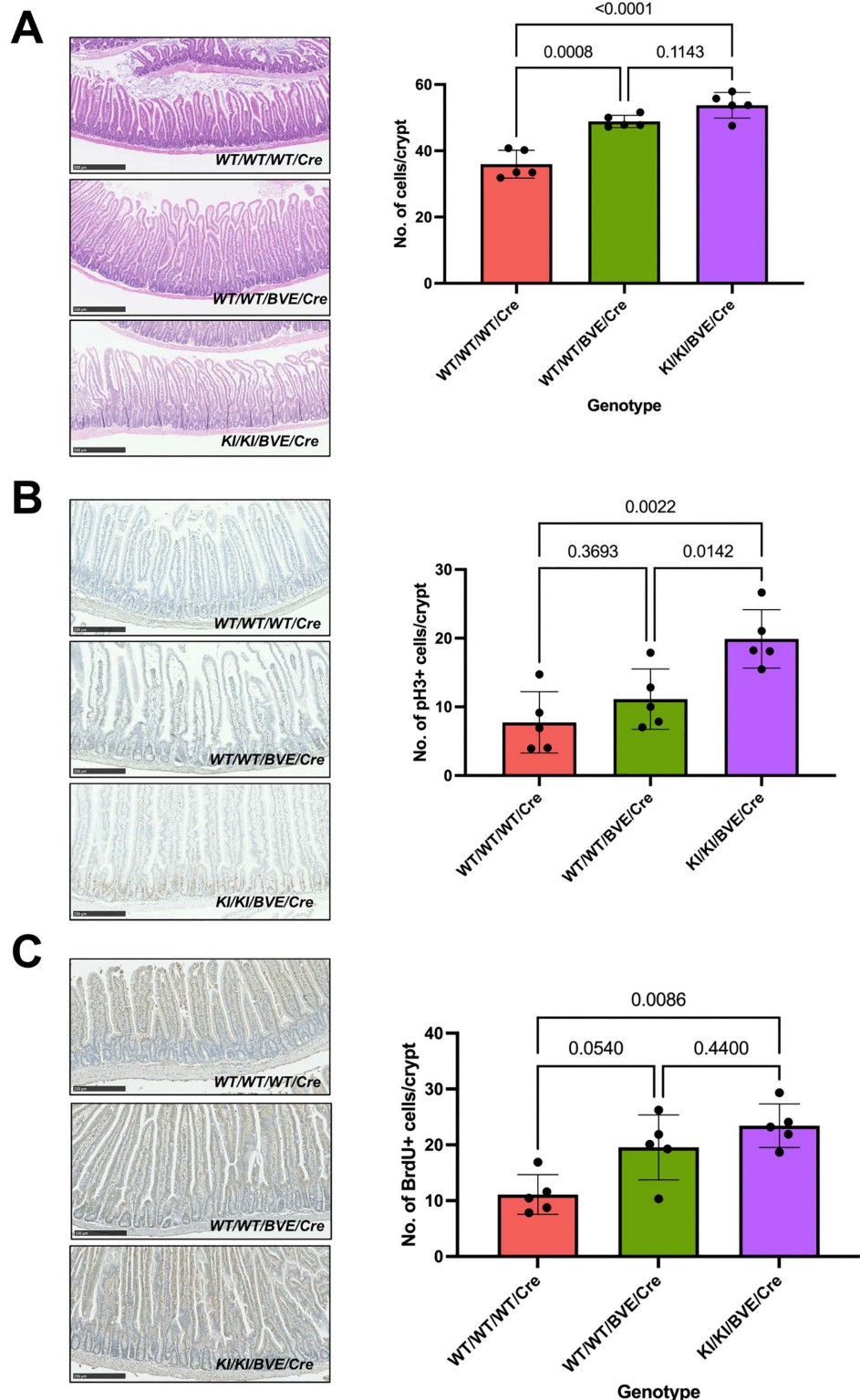

**Fig 2. Effect of genotype on crypt hyperplasia.** (A) Examination of crypt length following $^{V600E}Braf$ expression in H&E-stained sections of tissue from the small intestine of WT/WT/WT/Cre, WT/WT/BVE/Cre and KI/KI/BVE/Cre mice. Intestinal tissue was obtained from animals that were injected with TM and terminated 3 days after the last TM injection. Scale bars = 500 µm, 10x magnification. The graph on the right shows crypt cell number in mice of each

genotype after 3 days of TM induction. 50 full crypts from intestinal tissue of 5 mice of each genotype were subjected to crypt cell counting. Data show mean±SD. Significance was determined by the Wilcoxon matched pairs test and p values are indicated. (B) Immunohistochemical staining for pH3 in intestinal tissue from the small intestine of WT/WT/WT/Cre, WT/WT/BVE/Cre and KI/KI/BVE/Cre mice. Intestinal tissue was obtained from animals that were injected with TM and terminated 3 days after the last TM injection. Scale bars=250 µm, 10x magnification. The graph on the right shows quantitation of the number of pH3 positive cells per crypt in intestinal tissue of mice of each genotype. 50 full crypts from intestinal tissue of 5 mice of each genotype were subjected to crypt cell counting. Data show mean±SD. Significance was determined by the Wilcoxon matched pairs test and p values are indicated. (C) Immunohistochemical staining for BrdU incorporation in intestinal tissue from the small intestine of WT/WT/WT/Cre, WT/WT/BVE/Cre and KI/KI/BVE/Cre mice. Intestinal tissue was obtained from animals that were injected with TM and terminated 3 days after the last TM injection. Mice were injected with BrdU 3 hours prior to termination. Scale bars=250 µm, 10x magnification. The graph on the right shows quantitation of the number of BrdU positive cells per crypt in intestinal tissue of mice of each genotype. 50 full crypts from intestinal tissue of 5 mice of each genotype were subjected to crypt cell counting. Data show mean±SD. Significance was determined by the Wilcoxon matched pairs test and p values are indicated.

core regulatory genes of Wnt signalling (Fig 4C). However, this transcriptional regulation of Wnt pathway components as well as target genes was independent of *Gsk3αβ* status (Fig 4B, C)

## Discussion

GSK3αβ consists of two highly conserved isoforms, GSK3α and GSK3β, that are multifunctional serine-threonine kinase members of the CMGC (cyclin-dependent kinase [CDK], mitogen-activated protein kinase [MAPK], glycogen synthase kinase [GSK3], CDC-like kinase [CLK]) group of kinases [32]. They are both widely expressed and share 98% identity in their kinase domains. GSK3α and β isoforms are activated by tyrosine phosphorylation in their T-loop activation domains (Y279 and Y216, respectively) but inactivated by serine phosphorylation at their N-termini at serines 21 and 9 respectively [33]. Inactivation of GSK3αβ is considered to be critical for its role in insulin signalling and glycogen metabolism [11,34,35]. Besides this well characterised function, GSK3αβ has also been shown to play a role in various signalling pathways involved in the control of cell death, proliferation, differentiation and motility [36,37]. Linked to this, GSK3αβ has been implicated in numerous pathologies including cancer, cardiovascular and neurological diseases [38,39]. The role of GSK3αβ in cancer is incompletely understood with some studies suggesting pro-tumorigenic functions and other showing anti-tumorigenic functions [20]. Over-expression and increased activity of GSK3αβ has been detected in various cancer types including CRC [40,41]. Indeed, the effectiveness of GSK3β inhibitors in suppressing tumour growth in patient derived xenografts of CRC in mouse models has raised the possibility of these inhibitors being used as therapeutic agents in patients [41]. However, in contrast, other studies have identified GSK3αβ to be a tumour suppressor by inhibiting proliferation, with the best characterised function of GSK3αβ being as a suppressor of the Wnt signalling pathway through regulation of the destruction complex that targets β-catenin for degradation [42].

Studies of the insulin signalling pathway have shown that N-terminal phosphorylation of GSK3αβ by PKB are associated with inactivation because they cause the GSK3 N-termini to associate with the substrate binding pocket, preventing the binding of primed substrates [11,43]. Indeed, many signalling pathways appear to converge at this node of GSK3αβ inactivation, which is targeted by different kinases in response to a specific physiological signal [21,44]. For example, activation of the MAPK pathway by growth factors such as EGF induces GSK3αβ inactivation via N-terminal serine phosphorylation by p90[RSK] [45]. Activation of p70[S6K], PKA and PKC have also been associated with GSK3αβ inactivation via N-terminal phosphorylation in response to a variety of signals [44]. By contrast, inactivation of GSK3αβ in the context of the Wnt pathway is known to be mediated by Axin within the destruction complex rather than serine phosphorylation [46]. Through our own studies we have detected increased phosphorylation of GSK3α/β serines 21/9 in the mouse intestine following expression of the common oncogene [V600E]*Braf* and an increase in the expression of Wnt signalling pathway target genes [7,9]. Here, we have sought clarity as to the functions of N-terminal serine phosphorylations of GSK3αβ in Wnt pathway regulation and tumour development on the [V600E]*Braf* background.

Our data show that the homozygous *Gsk3αβ* knock-in mutations do not influence mouse survival, tumour burden or tumour grade following long-term expression of [V600E]*Braf* (Fig 1). There is a marginal effect on crypt hyperplasia

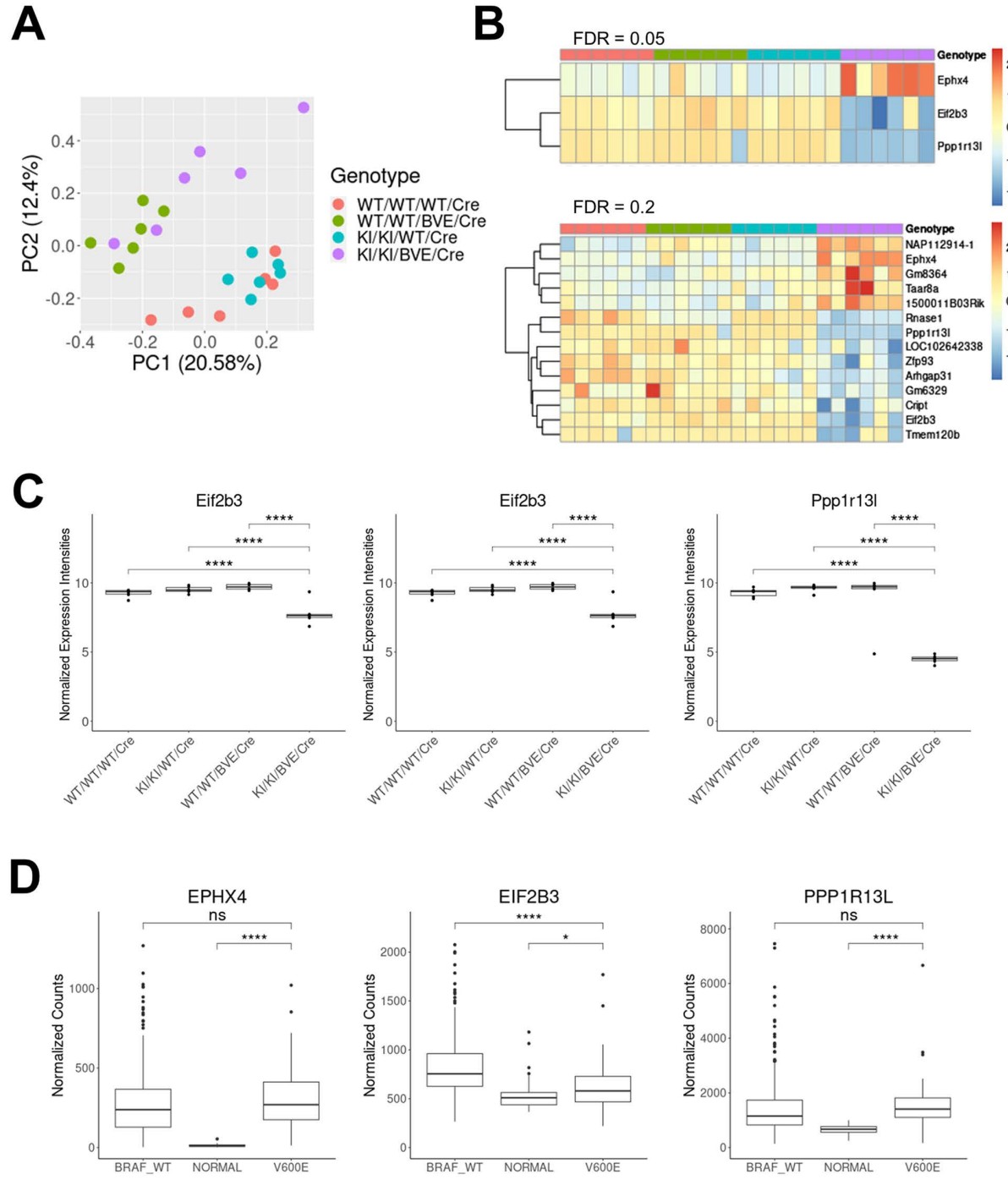

**Fig 3. Transcriptomic analysis of intestinal tissue.** (A) Principal component analysis of control and KI/KI mice showing clustering by genotype (n = 6 mice per group). (B) Heat map of differentially expressed genes at FDR 0.05 (top) and 0.2 (bottom). (C) Box plots showing individual values for the normalised expression of the 3 differentially expressed genes in KI/KI/BVE/Cre mice *Ephx4*, *Eif2b3* and *Ppp1r13l*. Significance was assessed using t-test, ****p < 0.0001. (D) Box plots showing *Ephx4*, *Eif2b3* and *Ppp1r13l* expression in the human TCGA dataset comparing normal tissue to tumour samples. Tumours were split according to their *BRAF* mutation status. P values are indicated, *p < 0.05, ****p < 0.0001.

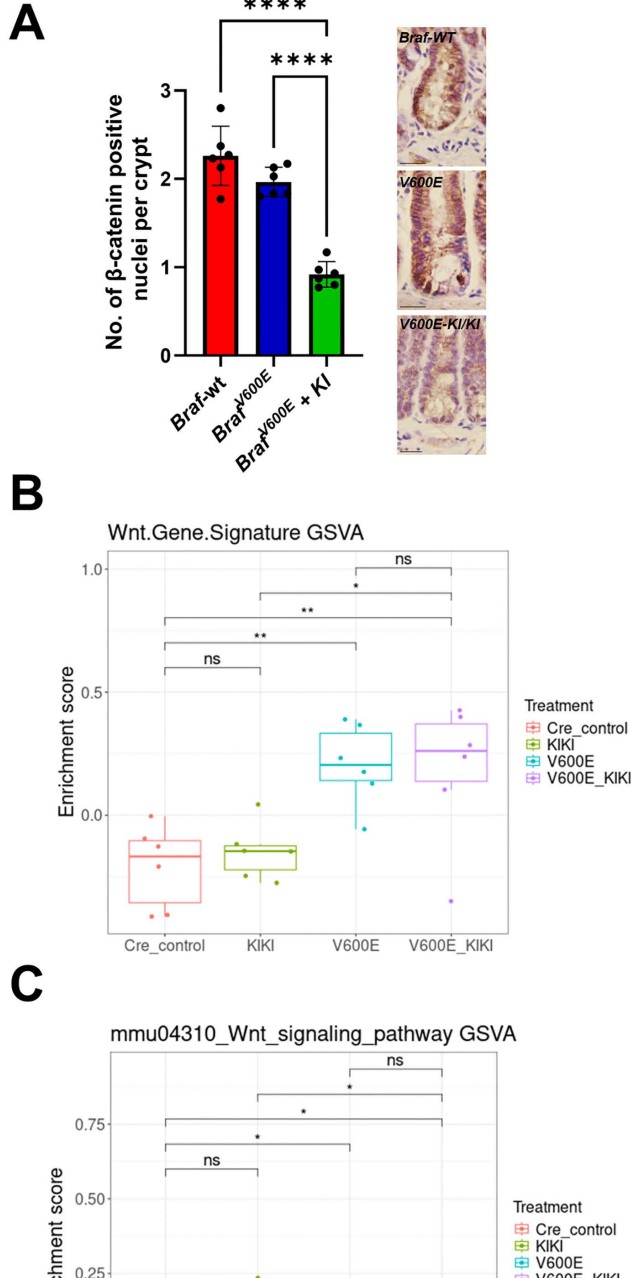

**Fig 4. Regulation of Wnt signaling.** (A) Quantification of nuclear β-catenin in intestinal tissue isolated from mice of the indicated genotypes. Examples of stained crypts are shown. 30 full crypts from intestinal tissue of 6 mice of each genotype were subjected to crypt cell counting. Data are plotted as mean±SD and were analysed using one-way ANOVA with Tukey multiple comparison test. ****$p < 0.0001$. Scale bars=20 μm, 40x magnification. (B) GSVA score of a Wnt gene signature comprising Wnt target genes in mice of the indicated genotypes. (C) GSVA score of Wnt pathway component signature comprising the main regulatory genes in Wnt signalling in mice of the indicated genotypes.

following short term expression of $^{V600E}Braf$ with a slight increase in the number of mitotic cells per crypt as measured by phospho-histone H3 staining (Fig 2B), but integrity of intestinal tissue is retained intact (Fig 2A). Importantly, when we assessed β-catenin localisation and Wnt activity, we found conflicting results. On the one hand, we detected decreased accumulation of nuclear β-catenin in mice bearing the non-phosphorylatable *Gsk3αβ* mutations (Fig 4A). However, these changes were not mirrored by alterations in expression of Wnt signalling components or target genes at the transcriptional level (Figs 4B, C). The only significant changes to gene expression were associated with $^{V600E}Braf$, which triggered an upregulation of the Wnt gene signature and a concomitant downregulation of the Wnt pathway component signature (Figs 4B, C). These data are consistent with the mild phenotypic impact of *Gsk3αβ* knock-in mutations and suggest that the reduction in β-catenin nuclear localisation observed in the double knock-in tissue on the $^{V600E}Braf$ background is insufficient to impair Wnt pathway activity. However, it cannot be ruled out that the $^{V600E}Braf$ mutation uncouples Wnt activity from the axin-dependent regulation of β-catenin by GSKα/β. Overall, these data show that the inability to phosphorylate S21/9 of GSK3does not influence the physiological or tumorigenic effects of $^{V600E}$Braf. This is consistent with our previous work showing that phosphorylations of serines 21/9 of GSK3αβare not required for cell lineage commitment or Wnt signalling in the context of the normal mouse intestine [23].

Transcriptomic analysis is supportive of the conclusion that the GSK3αβ phosphorylation plays very limited roles in intestinal homeostasis and oncogene signalling with only 3 genes being further altered in the presence of the *Gsk3αβ* mutations in a statistically significant manner: *Ephx4*, *Eif2b3* and *Ppp1r13l*. Expression of *Eif2b3* and *Ppp1r13l* is decreased by 1.5-2-fold in the presence of the *Gsk3αβ* mutations whereas *Ephx4* expression is increased by 1.5-fold. Therefore, despite these changes being statistically significant, the overall effect size of the changes is relatively small and therefore their functional significance may be marginal. All three genes have potential roles in cancer based on their previously reported functions. *Ephx4* is a member of the Epoxide Hydrolase (EH) family, which includes Microsomal Epoxide Hydrolase (EH1, or mEH, encoded by *Ephx4* gene), Soluble Epoxide hydrolase (EH2 or sHE, encoded by *Ephx4* gene) and Epoxide Hydrolase 3 (EH3, encoded by the *Ephx3* gene). These genes are involved in eicosanoid metabolism and detoxification, which has previously been linked with cancer development [47,48]. *Ppp1r13l* is also known as Inhibitor of Apoptosis Stimulating Protein of p53 (*iASPP*) and codes for an evolutionary conserved regulator of p53 and inhibitor of p53-induced apoptosis [49]. Despite *iASPP* being considered a *bona fide* oncogene [50], recent data in pancreatic cancer showed that loss of *iASPP* accelerates tumour progression [51], suggesting that the role of *iASPP* in cancer is context dependent. *Eif2b3* codes for a subunit of the translation initiation factor eIF2B, which catalyses the exchange of GDP for GTP in the eukaryotic initiation factor 2 (eIF2). Its downregulation is interesting in the light of the role of protein translation in the aetiology of CRC [52]. Notably, the expression of all 3 genes is significantly upregulated in human CRC (Fig 3D), although no correlation with survival and *BRAF* mutational status was identified.

In conclusion, despite the robust induction of phosphorylation of serines 21/9 in GSK3 in the mouse intestine following expression of $^{V600E}Braf$, this event does not play a role in tumour initiation or progression or in the homeostasis of the intestinal tract. Therefore, our data indicate that targeting of GSK3αβ N-terminal phosphorylation is not a worthwhile therapeutic strategy for the treatment of $^{V600E}BRAF$-driven CRCs.

## Supporting information

**S1 Fig. Schematic of the mating strategy.** A) Mating strategy used for the generation of *GSK3α*$^{S21A/S21A}$; *GSK3β*$^{S9A/S9A}$; *Braf*$^{+/+}$; *VilCreER*$^{T+/0}$ and *GSK3α*$^{S21A/S21A}$; *GSK3β*$^{S9A/S9A}$; *Braf*$^{LSL-V600E/+}$; *VilCreER*$^{T+/0}$ mice. B) Western blot analysis showing abolished phosphorylation of GSK3 proteins in intestinal tissue isolated from double knock-in animals. ERK2 was used as loading control.
(TIF)

**S2 Fig. Impact of mutation S21A and S9A of GSK3ab phosphorylation.** Full uncropped western blot scans used for S1 Fig.
(TIF)

## Acknowledgments

We are extremely grateful to Dario Alessi for providing the *Gsk3αβ* mutant mice. We are also indebted to the Department of Biomedical Services, Preclinical Research Facility at the University of Leicester, for technical support and the care of experimental animals and the Genomic Services within Core Biotechnology Services at Leicester for help with RNA sequencing. We thank Dr Kees Straatman at Advanced Imaging Facility (AIF) at the University of Leicester. Furthermore, we acknowledge Mr David Brown for all mouse genotyping carried out for this project. We acknowledge Servier on Bioicons (www.bioicons.com) for the realization of S1A Fig.

## Author contributions

**Conceptualization:** Catrin A Pritchard.

**Data curation:** Nicolas B Sylvius.

**Formal analysis:** Alessandro Rufini, Pooyeh Farahmand, Paulina Rzasa, Kevin West, Nicolas B Sylvius, Catrin A Pritchard.

**Funding acquisition:** Alessandro Rufini, Catrin A Pritchard.

**Investigation:** Pooyeh Farahmand, Fiona Hey, Susan Giblett, Hong Jin, Nicolas B Sylvius, Caleb Green.

**Methodology:** Pooyeh Farahmand.

**Supervision:** Alessandro Rufini, Catrin A Pritchard.

**Writing – original draft:** Alessandro Rufini, Catrin A Pritchard.

**Writing – review & editing:** Alessandro Rufini, Catrin A Pritchard.

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
