## [Decision Letter · Decision Letter 0]

22 Dec 2024

Dear Dr. Rufini,

Thank you for submitting your manuscript to PLOS ONE. After careful consideration, we feel that it has merit but does not fully meet PLOS ONE’s publication criteria as it currently stands. Therefore, we invite you to submit a revised version of the manuscript that addresses the points raised during the review process.

We look forward to receiving your revised manuscript.

Kind regards,

Divijendra Natha Reddy Sirigiri

Academic Editor

PLOS ONE

Journal Requirements:

"This work was funded by a Cancer Research UK Programme grant (Ref A13083). P.R. was employed on a grant from the Cancer Prevention Research Trust (Ref RM60G0817). We are extremely grateful to Dario Alessi for providing the GSK3 mutant mice. We are also indebted to the Department of Biomedical Services, Preclinical Research Facility at the University of Leicester, for technical support and the care of experimental animals and the Genomic Services within Core Biotechnology Services at Leicester for help with RNA sequencing. Furthermore, we acknowledge Mr David Brown for all mouse genotyping carried out for this project. "

"A.R., RM60G0817, Cancer Prevention Research Trust, https://www.cancerpreventionresearch.co.uk, this funder did not play any role in the study design, data collection and analysis, decision to publish, or preparation of the manuscript

C.A.P. Ref A13083, Cancer Research UK Programme grant, https://www.cancerresearchuk.org, this funder did not play any role in the study design, data collection and analysis, decision to publish, or preparation of the manuscript"

Reviewers' comments:

Reviewer's Responses to Questions

**Comments to the Author**

1. Is the manuscript technically sound, and do the data support the conclusions?

Reviewer #1: Partly

Reviewer #2: Partly

Reviewer #3: Yes

2. Has the statistical analysis been performed appropriately and rigorously?

Reviewer #1: Yes

Reviewer #2: Yes

Reviewer #3: Yes

3. Have the authors made all data underlying the findings in their manuscript fully available?

Reviewer #1: Yes

Reviewer #2: Yes

Reviewer #3: Yes

4. Is the manuscript presented in an intelligible fashion and written in standard English?

Reviewer #1: Yes

Reviewer #2: No

Reviewer #3: Yes

Reviewer #1: In this manuscript, the authors attempted to investigate the role of phosphorylation of GSK3α/β isoforms at residues S21/S9 in the context of Braf (V600E)-driven tumor development. A conditional knock-in mouse model with combined homozygous knock-in mutations of GSK3α (S21A) and GSK3β (S9A) on the Braf (V600E) was developed. The obtained data indicate that the homozygous GSK3 knock-in mutations do not influence mouse survival, tumor burden or tumor grade induced by Braf (V600E). Results of transcriptomic analysis in small intestinal tissue further indicate that GSK3α/β knock-in mutation has little to no effect in intestinal homeostasis and oncogene signaling and the variability among groups is principally driven by expression of mutant Braf. The interpretation of results is sound. There are some issues that should be incorporated:

1. The phosphorylation levels at serines 21/9 in GSK3α/β and their downstream signaling transductions in the mouse intestine should be determined.

2. The resolution of the images should be improved.

Reviewer #2: 1. This study is a follow-up to the authors' previously published research. It is advisable to utilize a graphical representation to demonstrate the process of generating transgenic mice.

2. The description of GSK3 mutations should be altered to "constitutively active GSK3^S21/9A". In addition, there are various ways to describe GSK3α/3β in the manuscript; please make it consistent.

3. The location of intestinal tumors that exist in V600EBraf-expressing mice needs to be classified as either colorectal or small intestine.

4. The GSK3 isoforms have distinct roles in physiology. GSK3β knockout mice are non-viable due to induced liver cell death, while GSK3α knockout mice survive but have impaired spermatogenesis. GSK3α can partially compensate for GSK3β's role in β-catenin regulation when GSK3β is deleted. However, selective inhibition of GSK3β prevents β-catenin degradation, as GSK3α has a lower affinity for β-catenin. This requires further discussion.

5. The expression level of β-catenin needs to be investigated.

6. What is the PPI of Ephx4, Eif2b3, and Ppp1r13l?

7. Could transgenic mice have other physiological issues besides tumorigenesis?

Reviewer #3: The study makes a significant contribution to our understanding of V600EBraf-driven colorectal cancers (CRCs) by providing valuable insights into the role of GSK3α/β phosphorylation. It effectively utilizes a conditional knock-in mouse model to explore this interaction. The findings, while indicating that GSK3α/β S9A/S21A mutations have only a marginal effect on crypt proliferation in the short term and no significant impact on survival, tumor burden, or tumor grade in the long term, also suggest that targeting GSK3α/β phosphorylation may not be a viable therapeutic approach for V600EBRAF CRCs. However, the study could benefit from further mechanistic exploration of how the identified genes (Ephx4, Eif2b3, and Ppp1r13l) contribute to crypt hyperplasia.

The animal experiment is commendable in its straightforward design; however, there are still aspects that warrant attention to enhance the clarity and readability of the manuscript.

1. In the Materials and Methods section, the standards for the humane euthanasia of experimental animals and severe endpoints should be written descriptively rather than as a list.

2. During the establishment of transgenic mouse models, was genotyping or gene sequencing performed, or how was the correctness of the transgenic mouse genotype confirmed? (I do not mean the gene expression analysis in Figure 3).

3. The figure legends were misplaced in the results section

4. In Fig 1C-button panel and 1E-left panel images, the magnification should be indicated in the text or figure legends.

5. The magnification should be indicated in the text or figure legends for the H&E images on the left panel in Fig2A, B, and C.

**Do you want your identity to be public for this peer review?** For information about this choice, including consent withdrawal, please see our Privacy Policy

Reviewer #1: No

Reviewer #2: No

Reviewer #3: **Yes:** Ying-Ray Lee

---

## [Author Response · Author response to Decision Letter 1]

12 May 2025

All comments have been addressed in the attached rebuttal letter.

---

## [Decision Letter · Decision Letter 1]

5 Nov 2025

Phosphorylations of Serines 21/9 in Glycogen Synthase Kinase 3α/β are dispensable for V600EBRAF-driven premalignant tumour development in the mouse intestine

PONE-D-24-37427R1

Dear Dr. Rufini,

We’re pleased to inform you that your manuscript has been judged scientifically suitable for publication and will be formally accepted for publication once it meets all outstanding technical requirements.

Kind regards,

Divijendra Natha Reddy Sirigiri

Academic Editor

PLOS ONE

Additional Editor Comments (optional):

Reviewers' comments:

Reviewer's Responses to Questions

**Comments to the Author**

Reviewer #1: All comments have been addressed

Reviewer #2: All comments have been addressed

2. Is the manuscript technically sound, and do the data support the conclusions?

Reviewer #1: (No Response)

Reviewer #2: Partly

3. Has the statistical analysis been performed appropriately and rigorously?

Reviewer #1: (No Response)

Reviewer #2: Yes

4. Have the authors made all data underlying the findings in their manuscript fully available?

Reviewer #1: (No Response)

Reviewer #2: Yes

5. Is the manuscript presented in an intelligible fashion and written in standard English?

Reviewer #1: (No Response)

Reviewer #2: Yes

Reviewer #1: (No Response)

Reviewer #2: (No Response)

**Do you want your identity to be public for this peer review?** For information about this choice, including consent withdrawal, please see our Privacy Policy

Reviewer #1: No

Reviewer #2: No

---

## [Editor Report · Acceptance letter]

PONE-D-24-37427R1

PLOS One

Dear Dr. Rufini,

I'm pleased to inform you that your manuscript has been deemed suitable for publication in PLOS One. Congratulations! Your manuscript is now being handed over to our production team.

Kind regards,

on behalf of

Dr. Divijendra Natha Reddy Sirigiri

Academic Editor

PLOS One